# Acupuncture, counselling or usual care for depression and comorbid pain: secondary analysis of a randomised controlled trial

A Hopton,[1] H MacPherson,[1] A Keding,[1] S Morley[2]

▶ Prepublication history and additional material is available. To view please visit the journal (http://dx.doi.org/10.1136/bmjopen-2014-004964).

▶ http://dx.doi.org/10.1136/bmjopen-2014-005144

## ABSTRACT

**Background:** Depression with comorbid pain is associated with a poor response to various treatments. The objective in this secondary analysis was to determine whether patients reporting pain have different depression and pain outcomes over time in response to acupuncture, counselling or usual care.

**Methods:** Self-reported ratings of depression and pain from 755 patients in a pragmatic randomised controlled trial of acupuncture (302) or counselling (302) compared to usual care alone (151) are described and analysed using a series of regression models and analysis of covariance. Patient-reported outcomes of Patient Health Questionnaire (PHQ)-9 for depression, SF36 bodily pain and EQ-5D, all at baseline, 3, 6, 9 and 12 months.

**Results:** At baseline, 755 patients reported EQ-5D pain categories; 384 (50.9%) reported moderate-to-extreme pain. Controlling for baseline depression, a linear regression model showed that the presence of pain at baseline was associated with poorer depression outcomes at 3 months mean difference=−1.16, (95% CI 0.12 to 2.2). Participants with moderate-to-extreme pain at baseline did better at 3 months if they received acupuncture (mean reduction in Patient Health Questionnaire 9 (PHQ-9) from baseline=6.0, 95% CI 5.0 to 7.1 and a mean reduction in SF-36 bodily pain=11.2, (95% CI 7.1 to 15.2) compared to improvements for those who received counselling (4.3, 95% CI 3.3 to 5.4; 7.6, 95% CI 3.6 to 11.6) or usual care (2.7, 95% CI 1.50 to 4.0: 7.2, 95% CI 2.3 to 12.1). In comparison, no notable differences were seen between treatment arms within the no pain comparator group.

**Conclusions:** Patients with depression and pain at baseline recovered less well from treatment over 3 months than those with depression and no pain. Reductions in both depression and pain were most marked in the acupuncture group, followed by the counselling group and then the usual care group.

### Strengths and limitations of this study

- This study targets the under-researched area of how pain may impact on the outcome of different treatments for depression.
- The research questions were clearly defined to establish the prevalence of pain in a depressed population, to describe the demographic profile of depressed people in pain compared to those who are depressed and pain free, to determine the impact of pain on the treatment outcomes, and in turn determine the effect of treatment for depression on the level of pain reported in the months subsequent to receiving treatment for depression.
- As a substudy of a larger pragmatic trial, the findings are important for three reasons; first, the emphasis on external validity with findings that are generalisable to primary care patients with depression across the UK. The method of randomisation to treatment allocation is a feature that was maintained through the subgrouping into a 'pain group' and 'no pain group' to provide control for temporal effects and random factors that might influence the outcome of treatment.
- This study was a substudy of a larger trial which was not powered to detect differences between the subgroups of moderate to extreme pain and no pain.
- The use of the BDI–II outcome was not measured at 3 months.
- Changes to factors such as sleeplessness, energy loss from within the BDI–II could not be assessed at the primary outcome point.
- The name of the medication prescribed to participants and the frequency of use was not recorded at baseline or as follow-up data, therefore the impact of the treatment on the types of antidepressant and analgesic medication used or the frequency of use could not be established.

For numbered affiliations see end of article.

**Correspondence to**
Ann Hopton;
ann.hopton@york.ac.uk

## INTRODUCTION

Around 50–66% of patients who are depressed also report pain.[1] [2] Depression accentuates existing pain problems[3] adding to the emotional distress and poor sleep irrespective of whether the pain had a known

cause.[4] Increased pain symptoms may also represent a greater severity of depression rather than a poor prognostic factor.[5] In terms of treatment, comorbid pain and depression are present in two-thirds of depressed primary care patients who start antidepressant therapy.[6] Pain is known to interfere with routine activities in 42% of primary care patients prior to starting pharmacological treatment for depression[7] and has a strong negative impact on achieving remission.[8] Too frequently there is a focus on the treatment of either pain or depression. The use of complementary therapies alongside conventional treatments for depression is widespread; this uptake may be attributed to the perceptions of ineffectiveness to patients, the distress of continued symptoms and the unwanted side effects of medication.[9] Treatment outcomes may be enhanced by taking a holistic approach, and delivering treatment options that address both pain and depression together.[10]

A meta-analysis of 18 000 participants data from 25 high-quality trials has shown that acupuncture is an effective treatment option for several chronic pain conditions.[11] Acupuncture is also regularly provided by acupuncturists to treat depression[12] although not routinely available as a treatment option for depression within the UK's National Health Service (NHS).[13] Counselling for depression is widely available in primary care practices[14] however there is limited evidence for counselling compared to usual care as a treatment for patients with depression and a chronic physical health problem.[4 15] A recent randomised controlled trial of acupuncture or counselling provided to patients with depression found that both interventions significantly reduced depression at 3 months when compared to usual care.[15] Pain was a secondary outcome in this trial, but has not yet been reported, and it will be of interest to compare the treatment outcomes of patients who had comorbid pain and depression with those who had depression alone.

The primary aim of this study was to find out whether people with depression who are also in pain have better or worse depression outcomes than those without pain. In terms of objectives, we document the prevalence of pain in a depressed population, the demographic profile of patients in pain and the relationship between pain and depression. We then determine depression and pain outcomes following treatment with acupuncture or counselling when compared to usual care alone.

## METHODS
### Design
This research is a substudy nested within a three-arm randomised controlled trial of acupuncture or counselling as adjuncts to usual care compared to usual care alone.[15] Full details of the Acupuncture, Counseling, and Usual care for Depression (ACUDep) trial protocol are reported elsewhere.[15 16] While maintaining the integrity of the 2:2:1 randomisation allocation of the ACUDep trial,[16] for the purpose of this substudy participants were divided into two groups according to their response to the EQ-5D pain statements at baseline: I have no pain or discomfort; I have moderate pain or discomfort; I have extreme pain or discomfort. People who reported either moderate or extreme pain were considered together as the 'pain group', the remainder were assigned as a 'no pain comparator group'. A summary of the subgroup allocations and other demographic variables are presented in table 1.

### Participants
Recruited to the main ACUDep trial[15] were 755 patients aged 18 or over and diagnosed with depression or who had consulted for depression within the previous 30 months with a baseline score of 20 or above on the Beck Depression Inventory-II, which this scale classes as 'moderate' to 'severe' depression.[17] Patients were recruited from general practices in rural and urban areas of Yorkshire, County Durham and Northumberland. Bodily pain scores on the SF-36[18] and any chronic physical conditions or pre-existing illnesses were reported by patients at baseline. Patients' details were recorded by the York Trials Unit and randomised by computer-generated block randomisation, with block sizes of 5 and 10 by an investigator with no clinical attachment to the trial. An unequal allocation ratio of 2:2:1 to acupuncture, counselling and usual care provided groups of 302, 302 and 151, respectively.

### Interventions
Within the main ACUDep trial[15] participants allocated to the acupuncture and counselling groups received the offer of up to 12 sessions on a weekly basis. The acupuncture intervention was performed according to a treatment protocol developed and agreed by acupuncture practitioners, which allowed for individualised treatment within a standardised theory-driven framework. The acupuncturists were members of the British Acupuncture Council with a minimum postqualification experience of 3 years, specific details of the intervention are published elsewhere.[19] The treatment protocol for the counselling intervention was based on competences independently developed for Skills for Health within National Occupational Standards for Psychological Therapies.[20] The counsellors were members of the British Association for Counselling and Psychotherapy with a minimum of 400 h postqualification experience and used a non-directive approach to help clients express feelings, clarify thoughts and reframe difficulties. Usual care available to all participants throughout the trial included prescribed pharmaceutical and other interventions provided by NHS primary or secondary mental health services.

### Outcome measures
The pain and depression outcome measures used for this substudy were collected as part of the ACUDep trial.[15] These included the primary depression score as

**Table 1** Demographics and variables of interest at baseline

| Characteristic | No pain N=371 (49%) | Moderate–extreme pain N=384 (51%) | Total N=755 (100%) |
|---|---|---|---|
| Age | | | |
| Mean (SD) | 39.9 (11.58) | 46.83 (14.11) | 43.5 (13.37) |
| Median (mininun, maximum) | 39 (18, 75) | 46 (18, 93) | 43 (18, 93) |
| IQR (25%, 75%) | 31, 48 | 38, 56 | 33, 53 |
| Missing | — | — | — |
| Sex | | | |
| Male | 88 (23.7%) | 113 (29.4%) | 201 (26.6%) |
| Female | 283 (76.3%) | 271 (70.6%) | 554 (73.4%) |
| Missing | — | — | — |
| Age left education | | | |
| Mean (SD) | 18.3 (3.90) | 17.53 (4.61) | 18.0 (4.37) |
| Median (minimum, maximum) | 17.0 (13, 52) | 16 (14, 54) | 16 (13–54) |
| IQR (25%, 75%) | 16, 21 | 16, 18 | 16, 19 |
| Missing | — | — | — |
| Employment | | | |
| Full-time education | 13 (3.5%) | 10 (2.6%) | 23 (3.1%) |
| Working full-time | 167 (45%) | 114 (29.7%) | 281 (37%) |
| Working part-time | 83 (22.4%) | 61 (15.9%) | 144 (19.5%) |
| Unable to work | 19 (5.1%) | 76 (19.8%) | 95 (12.9%) |
| Looking after home | 41 (11.1%) | 42 (10.9%) | 83 (11%) |
| Retired | 14 (3.8%) | 51 (13.3%) | 65 (8.8%) |
| Other | 26 (7%) | 22 (5.7%) | 48 (6.5%) |
| Missing | 8 (2.1%) | 8 (2.1%) | 16 (2.1%) |
| Painful health or medical condition | | | |
| Current painful health or medical condition | 41 (12.7%) | 299 (77.9%) | 346 (45.8%) |
| Onset before depression | 32 (8.6%) | 216 (56.2%) | 248 (32.8%) |
| Missing | 2 (0.5%) | 4 (1%) | 6 (0.8%) |
| Type of health problem | | | |
| Musculoskeletal | 21 (6%) | 221 (58%) | 242 (32%) |
| Other | 27 (7%) | 81 (21%) | 108 (14%) |
| No health problem | 321 (87%) | 81 (21% | 402 (53%) |
| SF-36 bodily pain | | | |
| Mean (SD) | 76.97 (22.53) | 39.03 (22.5) | 57.6 (28.44) |
| Median (minimum, maximum) | 77.5 (12, 100) | 41 (0, 100) | 52 (0, 100) |
| IQR (25%, 75%) | 62, 100 | 22, 51 | 32, 84 |
| Missing | 3 (0.39%) | 1 (1.3%) | 4 (0.5%) |
| Depression | | | |
| In the past 2 weeks | 264 (71.2%) | 310 (80.7%) | 574 (76.1%) |
| Missing | 2 (0.5%) | 10 (2.6%) | 12 (1.6%) |
| Not first major episode | 240 (64.7%) | 273 (71.1%) | 513 (67.9%) |
| Missing | 108 (29.1%) | 82 (21.3%) | 190 (25.2%) |
| 4+ previous episodes | 162 (43.7%) | 227 (59.1%) | 389 (51.5%) |
| Missing | 133 (35.8%) | 113 (29.4%) | 246 (32.6%) |
| Age at first major depressive episode | | | |
| Mean (SD) | 24.89 (11.56) | 25.47 (12.9) | 25.2 (12–28) |
| Median (minimum, maximum) | 21 (6, 72) | 23 (0–79) | 22 (0–79) |
| IQR (25%, 75%) | 16, 32 | 16, 31 | 16, 31 |
| Missing | 7 (0.9%) | 9 (1.2%) | 16 (2.1%) |
| Medication | | | |
| Depression medication in the past 3 months | 250 (67.3%) | 269 (70%) | 519 (68.7%) |
| Missing | — | — | — |
| Analgesic medication in the past 3 months | 114 (30.7%) | 245 (63.8%) | 359 (47.5%) |
| Missing | 4 (1%) | 2 (0.52%) | 6 (0.8%) |
| EQ-5D pain | | | |
| No pain | 371 (49.1%) | 0 | 371 (49.1%) |
| Moderate pain or discomfort | 0 | 298 (77.6%) | 298 (39.5%) |
| Extreme pain or discomfort | 0 | 86 (22.4%) | 86 (11.4%) |
| Missing | 3 | 4 | 7 (0.9%) |

Continued

**Table 1** Continued

| Characteristic | No pain N=371 (49%) | Moderate–extreme pain N=384 (51%) | Total N=755 (100%) |
|---|---|---|---|
| EQ-5D anxiety/depression | | | |
| Not anxious/depressed | 13 (3.5%) | 8 (2.1%) | 21 (2.8%) |
| Moderately anxious/depressed | 282 (76%) | 272 (70.8%) | 554 (73.4%) |
| Extremely anxious/depressed | 74 (19%) | 104 (27%) | 178 (23.6%) |
| Missing | 2 (0.5%) | 0 | 2 (0.3%) |
| EQ-5D mobility | | | |
| No problems walking about | 361 (31%) | 234 (31%) | 595 (79%) |
| Some problems walking about | 9 (1.2%) | 148 (19.7%) | 157 (20.8%) |
| Confined to bed | 0 (0%) | 1 (0.1%) | 1 (0.1%) |
| Missing | — | — | — |
| EQ-5D self-care | | | |
| No problems with self-care | 358 (47.5%) | 314 (41.7%) | 672 (89.2%) |
| Some problems with self-care | 11 (1.5%) | 68 (9%) | 79 (10.5%) |
| Unable to wash or dress | 1 (0.1%) | 1 (0.1%) | 2 (0.3%) |
| Missing | 1 (0.1%) | 1 (0.1%) | 2 (0.3%) |
| EQ-5D usual activities | | | |
| No problems performing usual activities | 251 (33.2%) | 111 (14.7%) | 362 (48.1%) |
| Some problems performing usual activities | 115 (15.3%) | 247 (32.8%) | 115 (48.1%) |
| Unable to perform usual activities | 5 (0.7%) | 24 (3.2%) | 29 (3.9%) |
| Missing | 2 (0.3%) | 0 (0%) | 2 (0.3%) |
| PHQ-9 | | | |
| Mean (SD) | 14.97 (5.15) | 17.0 (5.23) | 16.0 (5.29) |
| Median (minimum, maximum) | 15 (3, 27) | 17 (3, 27) | 16 (3–27) |
| IQR (25%, 75%) | 11, 19 | 13, 21 | 12, 20 |
| Missing | — | 1 (0.2%) | 1 (0.1%) |
| BDI–II | | | |
| Mean (SD) | 31.32 (8.29) | 33.6 (8.9) | 32.5 (8.72) |
| BDI–II group | | | |
| Moderate (20–28) | 155 (41.8%) | 129 (33.6%) | 284 (37.6%) |
| Severe (29–63) | 216 (52.8%) | 255 (66.4%) | 471 (62.4%) |
| Trial arm allocation | | | |
| Acupuncture | 156 (20.7%) | 146 (19.3%) | 302 (40% |
| Counselling | 151 (20%) | 151 (20%) | 302 (40%) |
| Usual care | 64 (8.5%) | 87 (11.5%) | 151 (20%) |
| Missing | — | — | — |
| Expectation of treatment allocated | | | |
| Very ineffective | 13 (3.5%) | 21 (9.6%) | 34 (4.5%) |
| Fairly ineffective | 39 (10.5%) | 57 (14.8%) | 96 (12.8%) |
| Cannot decide | 159 (42.9%) | 178 (46.4%) | 337 (44.9%) |
| Fairly effective | 122 (32.9%) | 87 (22.7) | 209 (27.8%) |
| Very effective | 38 (10.2%) | 37 (9.6%) | 75 (10.0%) |
| Missing | 0 | 4 (0.5%) | 4 (0.5%) |
| Treatment preference | | | |
| Acupuncture | 196 (52.8%) | 234 (60.9%) | 430 (57.5%) |
| Counselling | 94 (25.3%) | 70 (18.2%) | 164 (21.9%) |
| Usual care | 3 (0.8%) | 7 (1.8%) | 10 (1.3%) |
| No preference | 76 (20.5%) | 68 (17.7%) | 144 (19.3%) |
| Missing | 2 (0.5%) | 5 (1.3%) | 7 (0.9%) |

PHQ, Patient Health Questionnaire.

measured by the Patient Health Questionnaire (PHQ-9),[21] which scores each of the 9 DSM-IV criteria for depression as '0' (not at all) to '3' (nearly every day). The primary endpoint was at 3 months, with further follow-up at 6, 9 and 12 months. The patients' experience of the effect of the treatment received can also be expressed as depression-free days[22 23] which is an approximate summary measure derived from the PHQ-9 cut-off scores averaged over the period between measurements.[15] Patient reported pain measures were collected at baseline and at 3, 6, 9 and 12 months follow-up using the SF-36 Bodily Pain subscale, which has a range 0–100, where a score of 100 indicated no pain, and the EQ-5D categories of no pain or moderate pain or severe

pain. At baseline, patients also recorded their use of antidepressants and analgesic medication, and their preferences and expectations of treatment.

## Data analysis
### Prevalence and demographic profile of patients with pain
Descriptive statistics for EQ-5D pain and SF-36 pain scores were calculated. Based on the EQ-5D response categories, patients were allocated to a 'no pain' and 'moderate or extreme pain' group, using the median SF-36 cut-off where EQ-5D data were missing. A descriptive analysis of the demographic profile of the pain group and no pain comparator group was conducted: the participants' gender, average age of the sample population and pain groups population, their age on leaving education, the onset of their depression, the number of episodes they experienced, and the baseline scores of PHQ-9 and BDI–II baseline, SF-36 bodily pain score reported.

### Pain and depression at baseline
Analysis of variance (ANOVA) was used to compare the mean baseline PHQ-9 scores between the pain group and no pain comparator group. For the pain group alone, a Kendall's Tau correlation was used to test the association between the baseline scores of the PHQ-9 and SF-36 bodily pain score. This test was considered the most appropriate given the possibility of tied scores between the PHQ-9 and SF-36 bodily pain.

### Baseline pain and depression at 3 months
A series of regression models was applied to determine the influence of baseline pain in the presence of other demographic variables on the PHQ-9 depression outcome at 3 months, the primary endpoint of the trial. The first model predicted PHQ-9 at 3 months from the baseline EQ-5D Pain Group ('pain' or 'no pain'), controlling for baseline PHQ-9 scores. Additional predictors of PHQ-9 depression at 3 months were identified by individual univariate analyses of demographic variables, the BDI–II depression items and the five EQ-5D items while controlling for PHQ-9 scores at baseline. For this scoping exercise, a less conservative level of significance was set at $p < 0.1$ in order for potential covariates not to be missed and to maintain consistency with the methodology of the main ACUDep trial analysis. Any variables identified by univariate analysis were then included in a combined linear regression model, and any significant covariates were taken forward to the final model including pain grouping. Controlling for any remaining significant covariates of the final model ($p < 0.05$) and baseline PHQ-9 depression scores, analysis of covariance (ANCOVA) was used to test whether baseline pain affected treatment outcomes measured by the PHQ-9 score at 3 months.

Normality of continuous variables was assessed by inspection of histograms and Q-Q plots. All predictor variables included in the combined models were checked for collinearity in a correlation matrix and through assessment of tolerance and variance inflation factor (VIF) values.

### Baseline pain and treatment outcomes (depression and pain)
An interaction term between treatment and pain group in the above ANCOVA model was used to establish whether patients in pain responded differently to the treatments with regard to their PHQ-9 depression at 3 months than patients reporting no pain. The analysis was repeated for the outcome of depression-free days at 3 months, an approximate summary measure derived from PHQ-9 cut-off scores averaged over the period between measurements.[15 23] Descriptive analysis of the depression and pain scores was conducted over the 12-month follow-up period.

### Adverse events
A comparison of the adverse events reported between baseline and 12-month follow-up was conducted using proportions and OR.

### Missing data
Results of the main ACUDep analysis revealed an average response rate of 81% for the PHQ-9 at 3-month follow-up, and the primary analysis employed multiple imputation by chained regression using selected demographics and baseline pain and depression outcomes to impute PHQ-9 scores. For the present secondary analysis, imputation was not considered. However, for the initial allocation of patients into 'pain' and 'no pain' subgroups, the SF-36 bodily pain scale was used where the baseline EQ-5D pain item was not available in order to utilise all available outcome data. Outcomes were assumed to be missing at random, and the analysis of patients in their randomly allocated treatment groups (intention to treat) aimed to control for any random factors that might have influenced the outcome of treatment.

## RESULTS
### An exploration of pain and depression at baseline
#### What is the prevalence of pain?
Patients reported pain or discomfort on the EQ-5D questionnaire at baseline; 371 (49.1%) reported no pain or discomfort, 298 (39.5%) reported moderate pain or discomfort and 86 (11.4%) reported extreme pain or discomfort. The distribution of SF-36 bodily pain scores was negatively skewed; 140 (18.5%) patients reported a score of 100, indicating no pain at all, and the sample returned a median pain score of 52 (IQR 1=25, IQR 3=84).

### Allocation to 'pain' and 'no-pain' subgroups
For the purpose of this study, those reporting moderate and extreme pain or discomfort on the EQ-5D questionnaire were merged together to form a single 'pain

group' (n=384, 51%) with the remainder forming the 'no pain' comparator group (n=371, 49%). Patients who omitted to answer the EQ-5D pain question at baseline (n=7; 1%) were assigned to a group according to their SF-36 bodily pain score; three scored above the third IQR of the SF-36 bodily pain scale and were assigned to the no pain group, the remaining four patients at baseline scored below the median and were assigned to the pain group. Within the defined pain and no-pain subgroups, trial arm allocations retained the 2:2:1 ratio of the original ACUDep[15] randomisation process.

## What is the demographic profile of patients in pain and not in pain?

A summary of the demographic profiles, presented in table 1, show that the pain and the no-pain groups were comparable for most baseline variables. Notable exceptions were: The pain group members tended to be older than the no pain comparator group (mean of 47 years, vs 40 years) a difference which remained notable after temporarily removing 10 pain group members with outlying ages between 76 and 93 years. In terms of health and employment, 56% of the pain group (vs 9% of the no-pain group) reported a painful health condition or illness that predated the onset of depression, for which 64% (vs 31%) used analgesic medication regularly, 32% (vs 9%) were unable to work or retired.

## What is the relationship between pain and depression at baseline?

The baseline PHQ-9 depression scores indicate that the pain group reported higher levels of depression at baseline (mean PHQ-9=17.0, SD 5.2) than the no pain comparator group (mean PHQ-9=14.9, SD 5.2). Results of ANOVA confirmed the difference to be highly significant (mean difference 2.02, 95% CI 1.28 to 2.76). For the pain group alone the correlation between the PHQ-9 scores and SF-36 bodily pain scores was weak, but highly significant (Kendall's $\tau$ −0.172, $p<0.001$).

## Do patients reporting pain at baseline have different outcomes in response to treatment?
### Does baseline pain impact on depression at 3 months?

Using the average across all treatment groups, participants in the pain group showed less reduction in depression scores at 3 months compared to baseline (mean 16.70–12.06 at 3 months) than the no pain comparator group (baseline mean 14.06–9.10 at 3 months). A linear regression model found that the presence of moderate or extreme pain at baseline predicted a poorer outcome of depression treatment at 3 months (mean difference=−1.72, 95% CI 9.13 to 10.43, $p<0.001$), while controlling for baseline depression scores. A series of regression models (online supplement 1) identified three other significant predictors of poorer outcome of depression: poor EQ-5D mobility, loss of energy (BDI–II item 15) and being male. An ANCOVA model controlling for these covariates and baseline PHQ-9 revealed that the

effect of pain group remained significant, with patients with baseline pain having poorer depression outcomes (mean difference=−1.16, 95% CI −2.2 to −0.12, $p=0.028$).

Following inspection of summary statistics, histograms and Q–Q plots, approximate normality was ascertained for all continuous variables. When entering all covariates considered into a correlation matrix, all correlations were below 0.5 (range 0.022–0.446). For analysis models that included multiple covariates, all tolerance values were greater than 0.1 (range 0.788–0.991) and VIF values were less than 10 (1.010–1.269), suggesting no evidence for collinearity between predictors.

## Do depression scores change over time as a response to treatment for depression?

Figure 1 presents PHQ-9 depression scores by pain group and by trial arm at baseline and all follow-up time points. Controlling for baseline depression and covariates, an ANCOVA model including a pain group by treatment interaction term ($F_{(2603)}=2.138$, $p<0.119$) showed that in the pain group, participants showed a large reduction in depression with acupuncture at 3 months (mean reduction in PHQ-9 from baseline (6.0, 95% CI 5.07 to 7.11), with smaller reductions associated with counselling (4.3, 95% CI 3.3 to 5.4) and usual care (2.7, 95% CI 1.50 to 4.06). In comparison, no notable differences were seen between treatment arms within the no pain comparator group. Figure 1 shows that after the initial reduction in depression from baseline to 3 months, depression scores tended to remain relatively stable during the 6–12-month follow-up period in both the pain group and no pain comparator group.

In an ANCOVA model (controlling for baseline measures of depression, EQ-5D mobility, BDI–II loss of energy and sex) participants who received acupuncture reported significantly more depression-free days between baseline to 3-month follow-up than those receiving usual care alone (mean difference 15.2, 95% CI 3.12 to 15.05). The differences between counselling and acupuncture, and between counselling and usual care were not significant. The no pain comparator group also showed a trend in favour of acupuncture (see table 2).

## Does bodily pain change over time as a response to treatment for depression?

Using the SF-36 bodily pain score at 3-month follow-up as the endpoint and controlling for baseline SF-36 bodily pain and baseline PHQ-9 depression scores, results of ANCOVA show that the pain group continued to experience significantly worse pain after treatment for depression compared to the no pain comparator group (mean difference=14.57, 95% CI 9.73 to 19.40). There was also a significant interaction between pain group and treatment arm ($F_{(2,1)}=3.3$, $p=0.036$) where pain group patients who received acupuncture for depression experienced a greater reduction in SF-36 bodily pain (represented by an increase in scores) between baseline to 3-month follow-up (mean

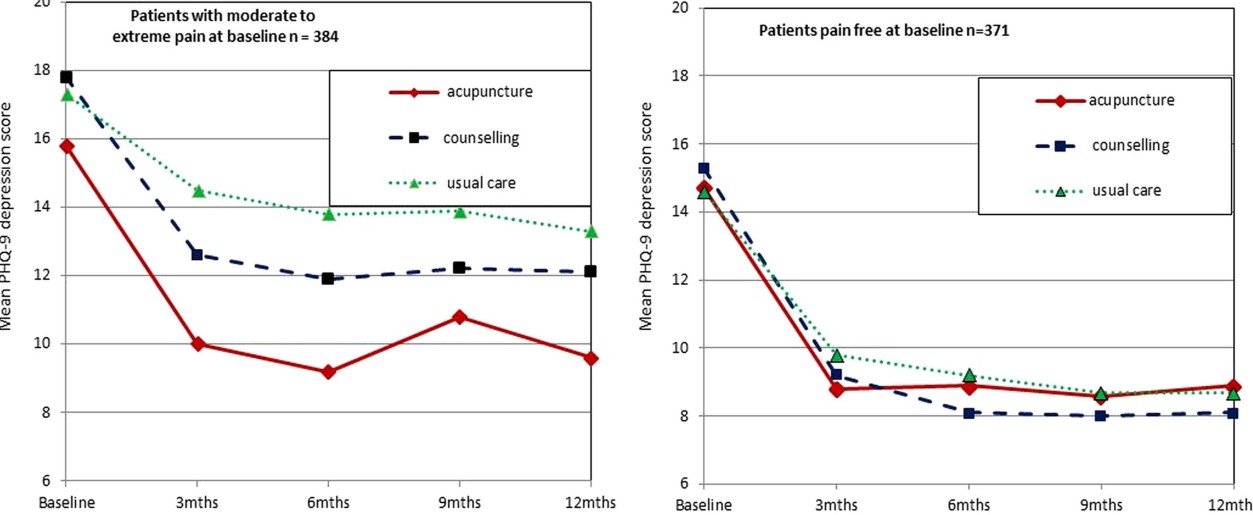

**Figure 1** Patient Health Questionnaire-9 depression scores by pain/no pain groups and trial arm at baseline and follow-up time points.

reduction=11.2, 95% CI 7.1 to 15.2), than those who received counselling (mean reduction=7.6, 95% CI 3.6 to 11.6) or usual care (mean reduction=7.2, 95% CI 2.3 to 12.1). The reduction in pain at 3 months persisted through to the 12-month follow-up point (figure 2), however, the median of pain group after 12 months (median=41, IQ1=31, IQ3=62) remained below the trial baseline median of 52 on the SF-36 bodily pain scale.

### Adverse events
Treatment response may also be moderated by adverse events; 32% of participants in the pain group reported some form of adverse event between baseline and 12 months compared to 15% of the no pain comparator group, and were twice as likely to report an adverse event (OR=2.05, 95% CI 1.47 to 2.88).

### DISCUSSION
### Summary of results
Participants with moderate-to-severe pain at baseline had worse outcomes for depression than the no pain comparator group in all three treatment arms after controlling for baseline depression. The results of this substudy confirm the main results of the ACUDep trial[15] by showing that at 3 months, both acupuncture and

counselling interventions were effective for depression compared to usual care alone irrespective of the presence of comorbid pain. The results also extend the findings of the ACUDep trial[15] by showing that acupuncture and counselling remain effective treatment options for patients with depression who also have comorbid pain. In addition, participants in the pain group had greater reductions in both depression symptoms with acupuncture from baseline to 3 months than those who received counselling or usual care. All treatment options were effective in reducing pain between baseline and 3-month follow-up after controlling for baseline pain, however, acupuncture delivered a greater degree of pain relief than counselling or usual care in the short-to-medium term.

### Strengths and limitations
This study targets the under researched area of how pain may impact on the outcome of different treatments for depression. The research questions were clearly defined to establish the prevalence of pain in a depressed population, to describe the demographic profile of depressed people in pain compared to those who are depressed and pain free, to determine the impact of pain on the treatment outcomes, and in turn determine the effect of treatment for depression on the level of pain reported in the months subsequent to receiving treatment for depression. As a substudy of a larger pragmatic trial, the findings are important for three reasons; first, the emphasis on external validity with findings that are generalisable to primary care patients with depression across the UK. Second, the method of randomisation to treatment allocation is a feature that was maintained through the subgrouping into a 'pain group' and 'no pain group' to provide control for temporal effects and random factors that might influence the outcome of treatment. Finally, the standardised treatment protocols allowed for

**Table 2** Summary of depression-free days at 3 months from baseline by pain group and trial arm

| Treatment received | Depression-free days | | |
| --- | --- | --- | --- |
| | No pain Mean (SD) | Moderate–extreme pain Mean (SD) | Total Mean (SD) |
| Acupuncture | 37.3 (26.4) | 31.3 (23.8) | 31.7 (25.3) |
| Counselling | 33.0 (22.2) | 20.7 (21.9) | 26.9 (22.9) |
| Usual care | 32.0 (23.5) | 16.1 (22.2) | 22.6 (23.5) |

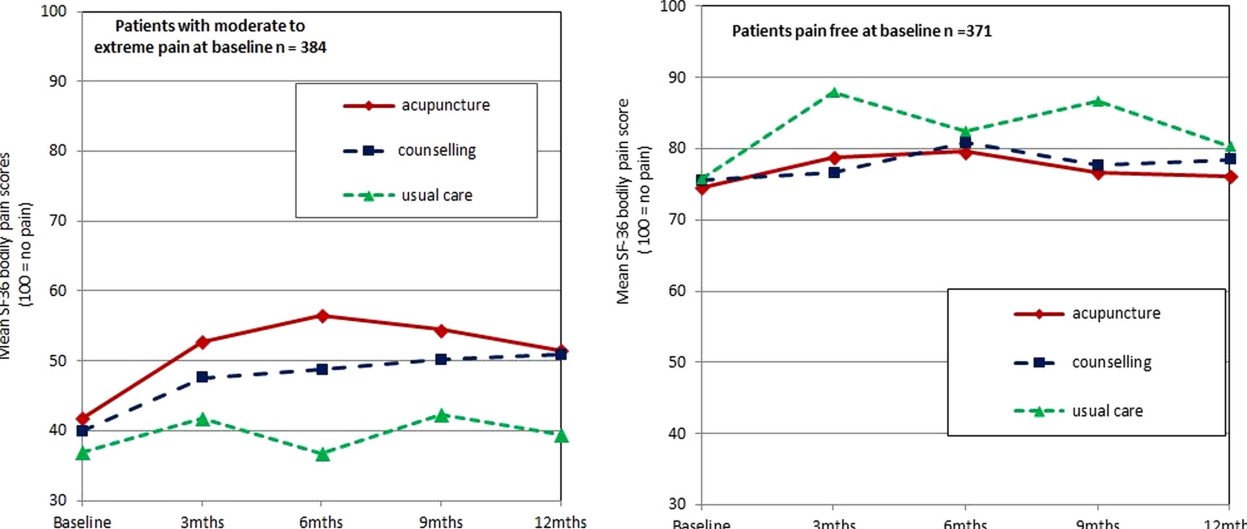

**Figure 2**  A summary of the SF-36 bodily pain scores by pain/no pain group and trial arm at baseline and follow-up time points.

treatment to be tailored to individual patient's needs regarding pain and depression while maintaining high standards of care, qualifications and practice experience. In terms of limitations, with regard to missing data; the patients who did not respond to follow-up at 3 months were divided equally between the pain group and no pain group, and their baseline levels of depression were similar to those who did respond. This study was a sub-study of a larger trial which was not powered to detect differences between the subgroups of moderate-to-extreme pain and no pain. The use of the BDI–II allowed for physical symptoms of depression to act as predictors to treatment outcome at 3-month follow-up, however this outcome was not measured again at 3 months. Therefore, changes to factors such as sleeplessness, energy loss from within the BDI–II could not be assessed; however, these attributes will be explored qualitatively in a follow-up paper. A further limitation is that the actual medication prescribed to participants and the frequency of use was not recorded at baseline or as follow-up data, therefore the impact of the treatment on the types of antidepressant and analgesic medication used or the frequency of use could not be established.

### Relationship to literature
#### Incidence of comorbidity
In terms of comorbidity, the estimated prevalence of moderate-to-extreme pain within our study population of depressed patients is 51%, which is comparable with the 50–66% in previous literature.[1 2 6 24] This is a consequence of the ACUDep trial[15] targeting primary care patients with moderate-to-severe depression on the BDI–II. The doubling of adverse events reported by patients in moderate-to-extreme pain may be accounted for by attendance for hospital investigations and overnight admissions in relation to existing illnesses. However, patients with comorbid pain and depression tend to

exhibit a cognitive bias specific to negative aspects of health[25] and are more likely to report less favourable outcomes of treatment.[26] Therefore, it is possible that treatment-related minor adverse events are more likely to be reported by patients who suffer from chronic pain.

#### Predictors of depression
Evidence from the English longitudinal study of ageing[27] identified pain and mobility disability at baseline as predictors of comorbid pain and depression. In a large European study, a higher number of pain locations, pain of the joints and longer duration of pain (for 90 days or more), daily use of pain medication and more severe pain at baseline were found to be associated with a significantly increased risk of still having a depressive or anxiety disorder after 2 years.[28] Together these factors are known to adversely affect the outcomes of treatment for depression.[7 8 29] Consistent with previous research, the majority of painful symptoms within the study sample were of musculoskeletal origin and accompanied by poor mobility and a loss of energy. Pain due to osteoarthritis is known to determine subsequent depressed mood through its effect on fatigue and disability.[30]

#### Reduction in pain
The reductions in pain from baseline to 3 months are within the lower range of 5–30 points for a minimally clinically important difference,[18 31] however, the majority of patients with pain remained below the trial baseline median of 52 on the SF-36 bodily pain scale at 3 months That patients reported reduced pain following acupuncture, is not surprising; 32% of patients had chronic muscular skeletal pain, for which there is a growing body of evidence supporting the efficacy of acupuncture.[11] With acupuncture being a holistic therapy, it is likely that acupuncturists incorporated treatment for

pain alongside treatment for the symptoms of depression. Patients who received counselling also reported a more gradual reduction in pain over the 12-month follow-up period. This finding is consistent with a Cochrane Review[31] which reports that psychological therapies, primarily CBT, can help people with people with chronic pain, reduce negative mood (depression and anxiety), disability and to a lesser degree reduce pain over a 6-month period.

### Implications for clinical practice and future research

Frequently patients with comorbid pain and depression will attribute their complaint to one or other of the conditions and seek help accordingly.[32] Patients presenting with pain or sleep disturbances are at risk of having a diagnosis of depression missed.[30] For treatment success, pain and depression should be recognised and treated from the outset. The evidence emerging for the current study is that both acupuncture and counselling appear to have the potential to reduce symptoms associated with pain and depression concurrently, potentially affording patients' relief from symptoms of depression and reduced intensity of pain in both the short and longer term. This study provides a platform from which to develop a larger investigation specifically designed with sufficient power to determine the impact of acupuncture or counselling when compared to other treatments for depression in patients with comorbid pain and depression.

### CONCLUSION

Patients who had moderate-to-extreme pain comorbid with depression at baseline recovered less well when compared to those who were pain free. Over 3 months, larger reductions in depression and pain scores were found in those who received acupuncture compared to those receiving counselling, and in turn these were greater than those receiving usual care. For those in pain at baseline, both acupuncture and counselling delivered a clinically meaningful reduction in bodily pain over 12 months.

**Author affiliations**
[1]Department of Health Sciences, University of York, York, UK
[2]Leeds Institute of Health Sciences, University of Leeds, Leeds, UK

**Acknowledgements** The authors would like to acknowledge the contribution of the patients, counsellors, acupuncturists and general medical practitioners; the ACUDep trial management team and members of MIND York.

**Contributors** AH conducted the analysis, interpreted the data, drafted and revised the article. HM critically revised important intellectual content and gave final approval for publication. AK provided substantial contribution to the conception and design and final approval for publication secondary analysis of the data and final approval for publication. SM provided substantial contribution to the conception and design, and final approval for publication.

**Funding** This article presents independent research funded by the National Institute for Health Research (NIHR) under its Programme Grants for Applied Research Programme (RP-PG-0707-10186).

**Competing interests** None.

**Ethics approval** Ethical approval was obtained within the ACUDep trial (ISRCTN63787732) from the York NHS Research Ethics Committee (ref:09/H1311/75).

**Provenance and peer review** Not commissioned; externally peer reviewed.

**Data sharing statement** No additional data are available.

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
