## [Reviewer comments · BMJ Open]

Some articles will have been accepted based in part or entirely on reviews undertaken for other BMJ Group journals. These will be reproduced where possible.

ARTICLE DETAILS

TITLE (PROVISIONAL)	ACUPUNCTURE, COUNSELLING OR USUAL CARE FOR DEPRESSION AND COMORBID PAIN: SECONDARY ANALYSIS OF A RANDOMISED CONTROLLED TRIAL
AUTHORS	Hopton, Ann; MacPherson, Hugh; Keding, Ada; Morley, Stephen

VERSION 1 - REVIEW

REVIEWER	Fowzia Ibrahim Clinical Trials Group Academic Department of Rheumatology King's College London
REVIEW RETURNED	18-Mar-2014

GENERAL COMMENTS	The statistical section, page 4, line 34. The choice of $p < 0.1$ could be explained in more details. Probably a univariate analysis was done and then any variable that had $p < 0.01$ taken forward to multivariate analysis were the significance level was set at 5%?. I would suggest a bit more explanation about the cut-off point would be helpful. Furthermore, the authors report median and interquartile ranges, probably some of the data is not normally distributed, it will be helpful to have a line in the method that explains some of the assumption, i.e. all continuous data were all normally distributed etc. Where they any variables that were collinear? It will be helpful to have coefficient, standard errors, 95% confidence interval and p-value for Supplement 1 univariate table, it is hard to read the F-statistics with degrees of freedom with p-value, this is measuring the fit of the model rather than showing the strength of association between the outcome and independent variables. Overall the results are well described but the statistical section need to be expanded, specially to assess whether there were collinearity and if the data was not normally distributed how this was dealt with. There was very missing data in each variable in the table one, but when this is combined it will probably have a greater impact, this is not in limitation or discussion, I would suggest to comment the impact the missing data could have on the results. Presumably the trial data used some sort of imputation methods to deal with missing data?
--

REVIEWER	Harm van Marwijk EMGO Institute of Health and Care Research VU University Medical Centre The Netherlands
REVIEW RETURNED	21-Mar-2014

GENERAL COMMENTS	This is a nice and relevant paper that is well-written. Their overall
---

	objective to determine whether patients reporting pain have different depression and pain outcomes over time in response to acupuncture, counselling or usual care is interesting. What they do is two subgroup analyses in a large trial, basically (pain/no pain). I do not think the idea to split the objective into several sub-objectives is presented convincingly as it is now, however. As such comparisons relate to different samples and require a different type of methodology (prevalence/prediction/RCT), this choice makes the paper more difficult to read. I suggest to present only the central objective and perhaps add the first one but not present them as steps in a (more or less) logical order. The first objective is linked to the last one, but the second one is separate, perhaps put that in a separate paper, or move it to the rear of the paper. The context of the trial is not easily assessed from this paper and needs more detail. How many patients participated in the original trial, for instance? Why did they think that acupuncture could be helpful for depression, etc. The abstract is clear about the trial results but the text is somehow less clear. The regression coefficients are not very informative for clinical readers: effect sizes would be better for instance, or to present means and SDs in a table perhaps. The figures are difficult to read. The authors seem to have restricted themselves to Anglo-Saxon sources. There are several large longitudinal studies that have recently looked at the relationship between depression, anxiety (!) and pain (See: Gerrits MM, et al. Impact of pain on the course of depressive and anxiety disorders. Pain. 2012 Feb;153(2):429-36.) Nice paper, would deserve publication.
--	---

VERSION 1 – AUTHOR RESPONSE

Reviewer 1

Comment: The statistical section, page 4, line 34. The choice of $p < 0.1$ could be explained in more details. Probably a univariate analysis was done and then any variable that had $p < 0.01$ (sic) taken forward to multivariate analysis were the significance level was set at 5%? I would suggest a bit more explanation about the cut-off point would be helpful.

Responses: Data analysis, paragraph 3: we have amended the text to include the following explanation:

Additional predictors of PHQ-9 depression at 3 months were identified by individual univariate analyses of demographic variables, the BDI-II depression items and the five EQ-5D items whilst controlling for PHQ-9 scores at baseline. For this scoping exercise, a less conservative level of significance was set at $p < 0.1$ in order for potential covariates not to be missed and to maintain consistency with the methodology of the main ACUDep trial analysis

Comment: Furthermore, the authors report median and interquartile ranges, probably some of the data is not normally distributed, it will be helpful to have a line in the method that explains some of the assumption, i.e. all continuous data were all normally distributed etc. Where they any variables that were collinear?

Response: We think the reviewer is referring to the data presented in table 1. The continuous data (e.g. age, PHQ-9) are presented showing the mean values, standard deviation, median and 1st and 3rd interquartile ranges. The data are presented in this format to maintain consistency with the data presented in the main ACUDep trial paper, and differences between mean and median values give a first indication regarding their distribution. The inspection of histograms and Q-Q plots showed that all

continuous outcomes were approximately normally distributed. When entering all covariates considered in this paper into a correlation matrix, all correlations were between 0.022 and 0.446 suggesting no evidence for collinearity between predictors. We have now included checks for normality and collinearity in the manuscript as detailed in response to the reviewer's question below. Comment: It will be helpful to have coefficient, standard errors, 95% confidence interval and p-value for Supplement 1 univariate table, it is hard to read the F-statistics with degrees of freedom with p-value, this is measuring the fit of the model rather than showing the strength of association between the outcome and independent variables.

Response: As advised by the reviewer, the univariate table in supplement 1 has been amended to show the coefficient, standard errors, 95% confidence intervals and p values.

Comment: Overall the results are well described but the statistical section need (sic) to be expanded, specially to assess whether there were collinearity and if the data was not normally distributed how this was dealt with.

Response: We have followed the reviewer's advice and expanded the methods and results section to include the assessment of normality and collinearity in the following additional text:

"Normality of continuous variables was assessed by inspection of histograms and Q-Q plots. All predictor variables included in the combined models were checked for collinearity in a correlation matrix and through assessment of tolerance and variance inflation factor (VIF) values."

"Following inspection of summary statistics, histograms and Q-Q plots, approximate normality was ascertained for all continuous variables. When entering all covariates considered into a correlation matrix, all correlations were below 0.5 (range 0.022 -0.446). For all models including multiple covariates, all tolerance values were greater than 0.1 (range 0.788 to 0.991) and VIF values were less than 10 (range 1.010 to 1.269) suggesting no evidence for collinearity between predictors."

Comment: There was very missing data in each variable in the table one, but when this is combined it will probably have a greater impact, this is not in limitation or discussion, I would suggest to comment the impact the missing data could have on the results. Presumably the trial data used some sort of imputation methods to deal with missing data?

Response: For the majority of baseline variables, very little (generally <2%) missing data was observed. The only instance of substantially missing data was for two follow-on questions for patients who indicated they had been depressed in the last two weeks. In the regression model including all covariates that were identified univariately, 23 participants (3%) with outcome data at 3 months did not have a complete set of predictors, so the impact was assessed to be minimal.

We have added the following paragraph on missing data in the data analysis section:

"Results of the main ACUDep analysis revealed an average response rate of 81% for the PHQ-9 at 3 months follow-up, and the primary analysis employed multiple imputation by chained regression using selected demographics and baseline pain and depression outcomes to impute PHQ-9 scores. For the present secondary analysis, imputation was not considered. However, for the initial allocation of patients into 'pain' and 'no pain' sub-groups, the SF-36 bodily pain scale was used where the baseline EQ-5D pain item was not available in order to utilise all available outcome data. Outcomes were assumed to be missing at random, and the analysis of patients in their randomly allocated treatment groups (intention to treat) aimed to control for any random factors that might have influenced the outcome of treatment."

We have also checked the profile of non-responders at three months follow up; equal numbers of the pain and no pain group were missing and their depression scores at baseline were not significantly different to those reported at baseline by the remainder of the sample. We have added the following text to the limitations section:

"with regards to missing data; the patients who did not respond to follow up at three months were divided equally between the pain group and no pain group, and their baseline levels of depression were similar to those who did respond".

Reviewer 2:

Comment: This is a nice and relevant paper that is well-written. Their overall objective to determine whether patients reporting pain have different depression and pain outcomes over time in response to acupuncture, counselling or usual care is interesting. What they do is two subgroup analyses in a large trial, basically (pain/no pain). I do not think the idea to split the objective into several sub-objectives is presented convincingly as it is now, however. As such comparisons relate to different samples and require a different type of methodology (prevalence/prediction/RCT); this choice makes the paper more difficult to read. I suggest to present only the central objective and perhaps add the first one but not present them as steps in a (more or less) logical order. The first objective is linked to the last one, but the second one is separate, perhaps put that in a separate paper, or move it to the rear of the paper.

Response: We appreciate the supportive comment and have taken the reviewers advice where possible. We have removed the split objectives from the paper; instead we present the results as a central objective. The reviewer is quite correct in saying that we use different methodologies of prevalence and prediction. To clarify; our first aim was to establish the presence of pain in a depressed population. Therefore to show that purpose we have amended the subheading of that section to read:

“An exploration of pain and depression at baseline”

With regards to the central section, we believe the reviewer is referring to the series of regression analyses. These were conducted to identify other demographic variables that might predict depression at the endpoint of three months. We have detailed this process in the data analysis section, but on the reviewers recommendation we have removed the section and placed it in the Web only supplement.

Comment: The context of the trial is not easily assessed from this paper and needs more detail. How many patients participated in the original trial, for instance? Why did they think that acupuncture could be helpful for depression?

Response: With regards to actual numbers and context or setting of the ACUDep trial we direct the reviewer to the Abstract results section, and Methods, participant section; in both sections we report the numbers of patients recruited to the main trial (n=755), the duration of their depression in the methods participants section we also describe the recruitment of patients from general practices in rural and urban areas of Yorkshire, County Durham and Northumberland

To address the question as to why acupuncture might be helpful, in the introduction section we have added the statement:

“The use of complementary therapies alongside conventional treatments for depression is widespread; this uptake may be attributed to the time taken for medications to show effect, which may suggest ineffectiveness to patients, whilst incurring the distress of continued symptoms and unwanted side effects of medication.”

Comment: The abstract is clear about the trial results but the text is somehow less clear. The regression coefficients are not very informative for clinical readers: effect sizes would be better for instance, or to present means and SDs in a table perhaps. The figures are difficult to read.

Response: As described above we have moved the regression models to the web only supplements. With regards to the clarity of the figures: the figures do show the clear differences in the trajectory of depression outcome over 12 months for both groups of patients and allow comparisons to be made between the two. With regards to the clarity: the figures are formatted to conform to the requirements of the BMJ Open to publish figures with 600x 600 dots per inch (dpi). We believe that the figures will be clear and easy to read when they are published in the online format.

Comment: The authors seem to have restricted themselves to Anglo-Saxon sources. There are several large longitudinal studies that have recently looked at the relationship between depression, anxiety (!) and pain (See: Gerrits MM, et al. Impact of pain on the course of depressive and anxiety disorders. *Pain*. 2012 Feb;153(2):429-36.)

Response: To address the point about Anglo Saxon sources, we have included a finding of the European study recommended by the reviewer, adding the following sentence in the Discussion section on the relationship to literature section related to predictors of depression:

“In a large European study, a higher number of pain locations, pain of the joints and longer duration of pain (for 90 days or more), daily use of pain medication, and more severe pain at baseline were found to be associated with a significantly increased risk of still having a depressive or anxiety disorder after 2 years”.